# Mating Competition between Wild and Artificially Reared Olive Fruit Flies

**Anastasia Terzidou** **, Dimitrios Koveos and Nikos Kouloussis ***

Laboratory of Applied Zoology and Parasitology, School of Agriculture, Thessaloniki University Campus, Aristotle University of Thessaloniki, 54124 Thessaloniki, Greece; anastasia.terzidou@gmail.com (A.T.); koveos@agro.auth.gr (D.K.)

\* Correspondence: nikoul@agro.auth.gr

**Abstract:** Mating success of artificially reared males of the olive fruit fly is essential for genetic control techniques for this pest. We studied the mating competition between males from a laboratory-adapted population reared with an artificial diet and wild males emerged from field-infested olives and grown in olives in the laboratory. We maintained virgin wild females or artificially reared females in cages together with virgin wild and artificially reared males and scored the percentages of different males in the mated pairs, mating latency, and mating duration. After mating, we determined the egg production and the size of spermathecae of females mated with different males. Our results indicate that artificially reared males are competitive to the wild males, and they mated in similar percentages with wild and artificially reared females. Mean mating latencies (SE) of wild females that mated with wild and artificially reared males were 69.8 (4.8) min ($n$ = 39) and 114.6 (8.1) ($n$ = 43) min, respectively. No difference was discovered in the mating duration or egg production between females that mated with a wild or artificially reared male. Wild females had higher spermathecae volume when they mated with wild males compared to artificially reared males (two-tailed $t$-test = −2.079, $df$ = 54, $p$ = 0.0423).

**Keywords:** *Bactrocera oleae*; spermathecae; competitiveness; laboratory-rearing

## 1. Introduction

The olive fruit fly (*Bactrocera oleae* Diptera, Tephritidae) is a major pest of olives and its distribution now covers the Mediterranean basin, North and Sub-Saharan Africa, southwest Asia and North America [1]. The developing larvae of the fly feed on the mesocarp of the fruit, causing more than 90% of crop losses if the population is not managed [2]. Insecticidal sprays are commonly used for the control of the olive fruit fly, but the result in-field was a developed resistance [3,4]. Biological control methods are also available, such as the release of parasitoid hymenopterans of the Braconidae family [5,6]. There are renewed efforts for the biotechnological management of the olive fruit fly, like the Sterile Insect Technique (SIT) and the Release of Insects carrying Dominant Lethal (RIDL) [7]. Both techniques require the rearing of insects of the pest population in large numbers in the laboratory, and subsequently, their release in the target area where they will mate with wild females and transfer their sperm (sterile or carrying a lethal gene). The quality and the ability of released males to search for and copulate effectively with wild females are crucial for the success of these techniques [8].

Studies on the mating behavior of olive fruit flies have indicated that colonization and rearing under laboratory conditions for many generations have a negative effect on male fitness and mating behavior [9,10]. Generally, long-term rearing under artificial conditions negatively affects behavioral and physiological aspects of insects [11], in addition to their longevity [12].

Additionally, sperm transfer and sperm storage are issues that must be considered for a successful SIT application. Spermathecae are the long-term sperm storage organs

in fruit flies, which can preserve the sperm until insemination. They are two sclerotized bodies, pear-shaped and black colored, connected to the dorsal part of the vagina through the spermathecal ducts [13]. Several factors may affect sperm transfer and storage, like age, size, nutritional status, and copula duration [14].

The aim of this study was to compare the mating success of wild (emerged from field-infested olives and grown in olives for 3 generations in our laboratory, thereafter referred to as W) and artificially reared flies (from a laboratory-adapted population and reared with an artificial diet for more than 500 generations, thereafter referred to as AR) with W and AR females in competitiveness tests. We measured the percentages of each male type that participated in mated couples, the mating latency, and the mating duration that followed. Secondly, we aimed to estimate the egg production of W and AR female flies that mated with W and AR males and the quantity of the transferred sperm by measuring the volume of their spermathecae.

## 2. Materials and Methods

### 2.1. Insect Rearing

The wild olive fruit fly colony was established with flies that emerged from infested olives (variety Megaritiki) collected in late September from the olive grove of the Aristotle University of Thessaloniki farm near the Macedonia airport of Thessaloniki (40.5382319776864, 22.995303221972858). Olive fruits collected in early August from the same farm were kept in the fridge (at 5–7 °C) and used for oviposition in our experiments. Emerged adults were maintained in colony wooden cages (30 × 30 × 30 cm) under laboratory conditions (24 ± 1.5 °C, RH 45 ± 5%, L:D 14:10), fed with a diet consisting of sugar, yeast hydrolysate, and water (ratio 4:1:5), and allowed to oviposit in olives. Water was provided with a soaked cotton stick extruding from a small water container. Flies of this wild population that were grown in their larval stages in olives for 2–3 generations in our laboratory were used in our experiments (W flies).

Additionally, we used AR olive fruit flies from the colony maintained in our laboratory for more than 20 generations. The colony was established from the "Democritus" strain, which was developed at the Democritus Nuclear Research Center, Athens, Greece and had been reared for more than 500 generations. Adult flies were kept in wooden cages (30 × 30 × 30 cm) and each cage contained about 200 individuals. Adult food was given in the form of a liquid diet consisting of sugar, yeast hydrolysate, and water (ratio 4:1:5) (no antibiotic was added). Egg yolk powder was added ad libitum as an extra protein source for the colony AR flies. They were allowed to oviposit on beeswax domes (diameter = 2 cm) (Figure S1) and eggs were collected every two days with a fine brush and washed with propionic acid solution (0.3%). The collected eggs were then placed directly on the larval diet inside a Petri dish (94 × 16 mm). The larval artificial diet consisted of 550 mL of tap water, olive oil (20 mL), Tween 80 (7.5 mL), potassium sorbate (0.5 g), nipagin (2 g), crystalline sugar (20 g), brewer's yeast (75 g), soy hydrolysate (30 g), hydrochloric acid 2N (30 mL), and cellulose powder as a bulking medium (275 g), as described in Tsitsipis et al. [15]. The diet was kept moist to stimulate last-stage larvae to exit the diet which, were then collected by sieving the sand on which the Petri dish was placed (Figure S2).

Newly emerged W and AR flies were separated by sex in the first 24 h of their emergence, kept in plexiglass cages (15 × 15 × 15 cm) that contained 20 flies each under the same conditions (T: 24 ± 1.5 °C, RH: 45 ± 5%, L:D 14:10) and fed with the same diet consisting of sugar, yeast hydrolysate, and water (ratio 4:1:5).

### 2.2. Male Mating Competitiveness Test

To study mating competitiveness, we maintained AR males, W males, and W females in plexiglass cages and scored mating percentages. More specifically, ten flies of each different rearing history were maintained in a plexiglass cage (15 × 15 × 15 cm), and mating percentages were determined from 16:00 until 21:00 (end of photophase). Before their transfer to the competition cages, AR and W male flies were anesthetized with $CO_2$

and painted on their thorax with a different nontoxic watercolor. When a successful mating occurred, the pair was removed, and another male of the same rearing history was added in the cage. Thus, the ratio of W males:AR males was maintained at 1:1. No new female fly was added, to ensure a maximum of 10 pairs per cage. The experiment was repeated on two different days with new sets of flies. There were 10 replications in each day, i.e., 10 cages with 30 flies (a total of 20 replications). For each pair, we scored the type of male and the duration of mating. Mating latency (time between the initiation of observations and the initiation of each mating) and mating duration were later calculated. The same bioassay was conducted to study mating competitiveness between AR and W males for AR females.

For the control, we maintained 20 W flies (10 females and 10 males) or 20 AR flies (10 females and 10 males) in each cage and scored mating duration and latency. In each treatment, there were 10 replications (10 cages with 20 flies).

According to Mazomenos [16] and Manoukas [17] and personal observations, sexual maturity occurs earlier in AR than in W flies. Meats et al. [18] discovered that sexual maturity begins on the 8th day of age in W flies of both sexes, while for AR males and females it begins on the 2nd day and the 3rd day of age, respectively. Therefore, in our experiments we used 7–8 and 12–13 day-old AR and W female flies, respectively.

Respective weights of W and AR flies of both sexes were measured before the mating experiment. Adult flies were anaesthetized with $CO_2$ and then placed on a precision balance (Kern & Sohn GmbH, D-72336, Balingen, Germany, model ALS 220-4, max 220 g, d = 0.1 mg).

Longevity of both sexes of W and AR flies has been estimated with previous bioassays in our laboratory, and is described in the Supplementary Material (Table S1).

### 2.3. Oviposition

After their mating with either a W or an AR male, females were maintained individually and allowed to lay eggs in olives for 10 days in transparent plastic cups of 400 mL volume, as described by Kouloussis et al. [19]. An olive fruit was placed on the bottom of each cage, to serve as an oviposition substrate. The olive fruit were examined under a stereoscope and the oviposition punctures or holes were scored. Each oviposition hole corresponds to an egg laid inside the fruit. For AR females, a wax dome was placed on the bottom of the cup and served as oviposition substrate. Every day, we scored the number of eggs laid in the olive fruit or the wax dome.

### 2.4. Spermathecae Volume

After oviposition, female flies were dissected under a stereoscope (Leica M28). W flies were dissected on the 22nd day of age and AR flies on their 17th day of age. Their spermathecae were photographed with a camera (Jenoptic Gryphax Naos), and using the camera software (GRYPHAX version 2.1.0.724), the lengths of the major and minor axis (DST1 and DST2) of each spermatheca vesicle were measured (Figure 1). Each vesicle was considered a spheroid [20] and its volume was estimated using Equation (1), where $r_1$ corresponds to the length of its semi-major axis and $r_2$ to the length of its semi-minor axis.

$$V_{spt} = \frac{4}{3} \pi r_1 (r_2)^2 \tag{1}$$

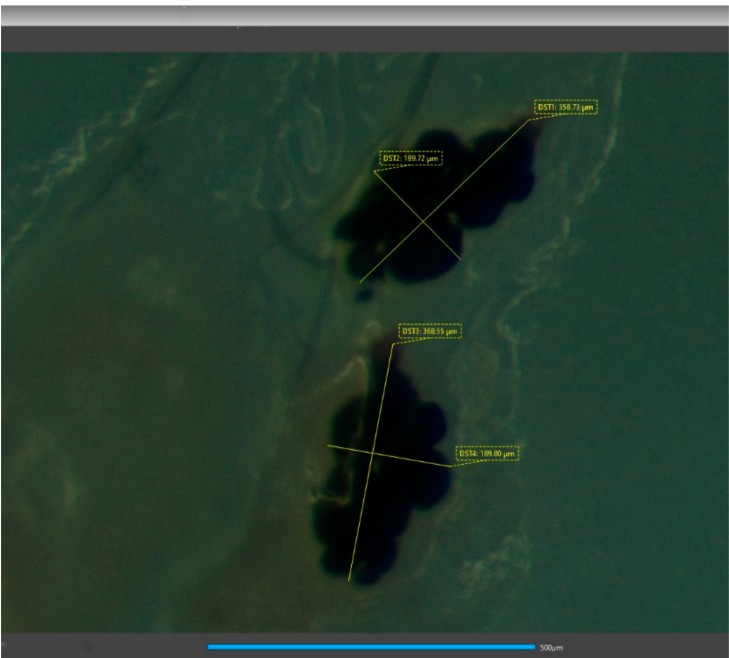

**Figure 1.** Spermathecae of wild females after mating with wild males. DST1 and DST2 correspond to the major and minor axis of the organ, respectively. Scale bar is equal to 500 μm. Photograph was taken with the camera Jenoptic Gryphax Naos and the measurements using its software (GRYPHAX version 2.1.0.724).

*2.5. Statistics*

For the male competitiveness tests, we scored the number of mated pairs, whether the male that participated was W or AR, and determined the relative number of pairs per replicate cage. Mean percentage of mated couples and standard error were calculated from the replicates. Deviation from randomness (1:1) was determined using a chi-squared goodness-of-fit test.

Mating latency and mating duration were compared among pairs of W females with W or AR males and pairs in their control groups. The same parameters were compared among pairs of AR females with W or AR males and their respective control group. One-way ANOVA followed by a Tukey post-hoc test was performed to compare mating latency and duration. Normality and homogeneity of variances of residuals were tested, and no serious violations were detected. The number of eggs and spermathecae volume between W females that mated with W or AR males were compared with a two-tailed *t*-test. The same parameters were compared between AR females that mated with W or AR males with a two-tailed *t*-test. Weights of W and AR flies of the same sex were also compared with a two-tailed *t*-test. For all analyses, the level of significance was a = 0.05. The statistical software package JMP 14.1.0 [21] and IBM SPSS Statistics 20 were used for all tests performed.

**3. Results**

*3.1. Male Competitiveness Test*

In the bioassay where we maintained W females with AR males and W males, it was discovered that AR and W males mated in similar percentages with W females ($\chi^2 = 0.016$, d$f = 1$, $p = 0.8993$). The mean percentage (SE) of participation of W males in the formed couples was 55.5 (5.5) % and that of AR males was 43.9 (5.5) %.

In the bioassay where we maintained AR females with AR males and W males, mating percentages between AR females and AR and W males did not differ significantly ($\chi^2 = 0.098$, d$f = 1$, $p = 0.7542$). The mean percentage (SE) of participation of W males in the formed couples was 43.4 (7.8) % and that of AR males was 56.1 (7.9) %.

The mean weight (SE) of W females was 6.67 (0.25) mg (*n* = 20) and of AR females was 6.82 (0.19) mg (*n* = 20). There was no statistical difference between them (two-tailed *t*-test = −0.454, *p* = 0.653, *n* = 20 for each type). Mean weight (SE) of W males was 5.65 (0.82) mg (*n* = 20) and of AR males was 5.47 (0.10) mg (*n* = 20). There was no statistical difference between them (two-tailed *t*-test = −0.454, *p* = 0.653, *n* = 20 for each type).

### 3.2. Mating Latency

One-way ANOVA among all treatments indicated statistical differences in mating latency (*F* = 19.415, *df* = 5, *p* < 0.0001) (Figure 2). When W females were allowed to mate with W and AR males, mean (SE) mating latency was significantly shorter in matings with W males [69.8 (4.8) min (*n* = 39)] than with AR males (114.6 (8.1) min (*n* = 43)) (Tukey post-hoc test, *p* = 0.0005). By contrast, when AR females were allowed to mate with W males and AR males, mating latency was 94.6 (11.4) min (*n* = 16) and 104.0 (8.7) min (*n* = 20), respectively, and did not differ significantly (Tukey post-hoc test, *p* = 0.9922). In the control, W males initiated mating with W females later than AR males with AR females [157.9 (6.9) min (*n* = 47) and 82.9 (9.1) min (*n* = 27) of mating latency, respectively]. Mating latency of W female—W male pairs was shorter when they mated in competition cages compared to the mating latency of W female—W male pairs without competition (control) (Tukey post-hoc test, *p* < 0.0001).

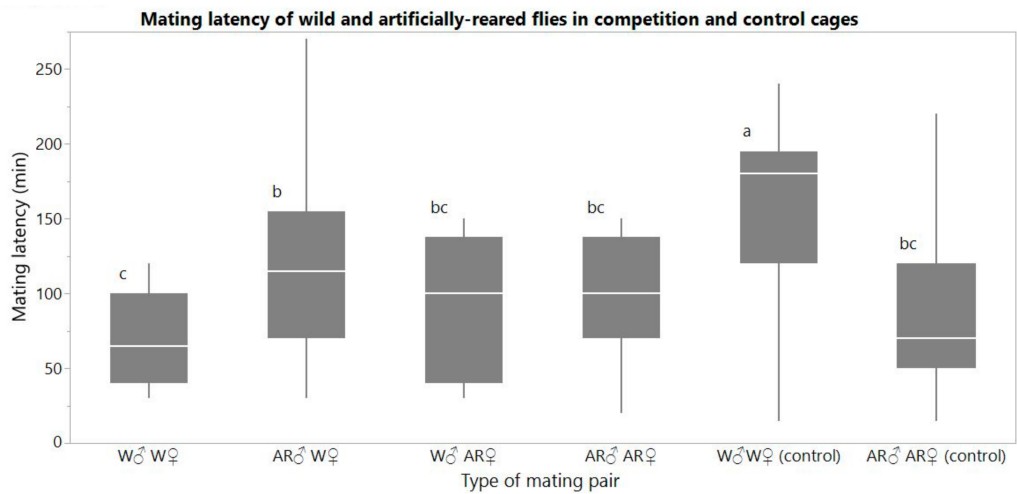

**Figure 2.** Boxplots of mating latency (min) of the pair combinations between wild and artificially reared flies in competition cages and in control cages. Different letters signify statistical difference (Tukey post-hoc test, *α* = 0.05).

### 3.3. Mating Duration

One-way ANOVA indicated that mating duration among all possible pair combinations and the control did not differ significantly (*F* = 0.8085, *df* = 5, *p* = 0.4917). In competition cages with W females, the mean (SE) mating duration with W and AR males was 173.9 (10.4) min (*n* = 39) and 167.3 (7.0) min (*n* = 43). Similarly, when AR females were allowed to mate with W and AR males, mating duration was 151.5 (13.6) min (*n* = 16) and 177.0 (12.2) min (*n* = 20), respectively. In control cages, the mean (SE) mating duration of W pairs was 168.0 (11.1) min (*n* = 47) and that of AR pairs was 160.9 (12.6) min (*n* = 27).

### 3.4. Egg Production and Size of Spermathecae

For W females, the mean number of eggs (SE) laid in olive fruits during the experimental period of 10 days was 52.8 (3.7) (*n* = 24) and 50.0 (2.8) (*n* = 24) when they had mated with a W male and an AR male, respectively. There was no statistical difference between the two means (two-tailed *t*-test = 0.593, *df* = 43, *p* = 0.556).

Similarly, for AR females, the mean number of eggs (SE) laid in wax domes during the experimental period of 10 days was 79.2 (5.2) eggs/female/10 days (*n* = 16) and 68.9 (6.5) eggs/female/10 days (*n* = 20) when they had mated with a W male and with an AR male, respectively. There was no statistical difference between the two means (two-tailed *t*-test = 1.181, *df* = 34, *p* = 0.224) (Table 1).

**Table 1.** Mean number of eggs/female (SE) laid during 10 days after mating and spermathecae volume (SE) (mm$^3$) of W and AR females mated with either W or AR males. Means in the same row followed by different letters differ significantly (two-tailed *t*-test, α level = 0.05).

| Variables | Female | Mated with W♂ | Mated with AR♂ | *p*-Value |
|---|---|---|---|---|
| Eggs/female/10 days | W♀ | 52.8 (3.7) a | 50.0 (2.8) a | 0.556 |
| Eggs/female/10 days | AR♀ | 79.2 (5.2) a | 68.9 (6.5) a | 0.224 |
| Spermathecae volume | W♀ | 0.0093 (0.0007) a | 0.0075 (0.0003) b | 0.0423 |
| Spermathecae volume | AR♀ | 0.0069 (0.0005) a | 0.0063 (0.0006) a | 0.870 |

However, AR females laid a significantly greater number of eggs than the W females (two-tailed *t*-test = −4.136, *df* = 55, *P* < 0.0001), irrespective of mating with W or AR males.

The mean volume (SE) of spermathecae of W females was significantly larger after mating with W males than AR males (two-tailed *t*-test = −2.079, *df* = 54, *p* = 0.0423) ,

The mean volume (SE) of spermathecae of AR females was similar after their mating with either a W or AR male (two-tailed *t*-test = −0.163, *df* = 34, *p* = 0.870) (Table 1).

The size of spermathecae of AR females was significantly smaller than the spermathecae of W females, irrespective of mating with a W or an AR male (two-tailed *t*-test = 2.948, *df* = 100, *p* = 0.004).

## 4. Discussion

Our experiments indicate that W and AR males of the olive fruit fly mated in similar percentages with W and AR females. Although our experiments were performed in cages under laboratory conditions, the results are encouraging and indicate that AR and W males are equally competitive for mating with W females. It is known that genetic methods of population suppression (like Sterile Insect Technique, SIT, or Release of Insects carrying Dominant Lethal gene, RIDL) [22,23] are based on the use of competitive male flies of laboratory-adapted strains. In the past, releases of sterile *B. oleae* males in the field demonstrated some success in suppressing the pest population in combination with a conventional pest management application [24]. A laboratory adapted strain of *D. ciliatus* (kept for more than 40 generations) indicated satisfactory male competitiveness compared with the wild flies [25]. In another tephritid fly named *Ceratitis capitata*, competitiveness tests in field cages with sterile/mass-reared and wild males indicated that sterile/mass-reared males mated with wild females at similar percentages as wild males, and their performance was improved when their diet was bacterially enriched. As noted by the authors, mass-reared sterile flies perform better if, prior to the bioassay, they are maintained in less crowded conditions than at the mass-rearing facilities [26]. Yet, sterile *Anastrepha ludens* males, from a mass-reared colony kept for more than 10 years, were able to mate with wild females in field cages and successfully compete with wild males, although they performance varied according to the origin of the wild population [27]. Our experiments were conducted in cages under laboratory conditions. There was no cage with fewer than 20% of mated females, which indicates that the experimental conditions were favorable and the tested insects were of the appropriate age and sexual maturation. However, bioassays performed in small cages may favor forced matings. Therefore, additional field experiments are required in order to verify our results.

Mating latency, used as a proxy for females' willingness to mate, was similar for AR female flies whether they mated with AR or W males. However, mating latency for W female flies was shorter when they mated with W than with AR males. We may assume that W males were courting more efficiently than the AR males, or were more successful

in male–male rivalry behaviors in the competition cages. In control cages, W flies mated later than AR flies. This is in accordance with earlier studies which indicated that mass-reared tephritid males tend to begin calling and mating earlier in the day than the wild population [28,29]. Mating latency is considered critical for the success of the SIT/RIDL, as earlier matings of wild males would potentially leave fewer opportunities for sterile males to transfer their sterile sperm to the wild females [11]. Additional bioassays in larger field cages and under natural light conditions are required to further explore differences in mating latency between AR and W flies.

Mating duration did not differ between any of the possible pair combinations in our experiments. In other mass-reared tephritid flies, such as *Ceratitis capitata*, shorter courtship and mating durations were observed compared to the wild population, and were attributed to adaptation to crowded rearing conditions [30]. However, shorter mating duration did not always result in smaller quantity of sperm storage or have any relation to the amount of sperm transferred to the female [31,32]. It is unclear which sex controls the duration of mating and it is difficult to differentiate between male and female influences [33].

The number of eggs laid by W or AR female flies did not depend on the type of male they mated with. However, the number of eggs laid was dependent on the female type, with AR flies laying more eggs than the wild ones (*t*-test, $p < 0.0001$). Although the oviposition substrate was different (olives vs. wax domes) and may have influenced the results, high oviposition is a favored characteristic during the mass-rearing selection process [34] and it was expected for AR female flies.

The spermathecae of W females that mated with W males were larger in volume compared to the spermathecae of W flies that mated with AR males (*t*-test, $p = 0.00423$). We estimated the spermathecae volume through their digital photographs and considered that it corresponds to the quantity of stored seminal liquid. In some hymenopteran species, a positive correlation was discovered to exist between the volume of the spermatheca and the number gametes it can store [35]. Collins et al. [36] have used digital photos of spermathecae of honeybee queens to calculate their volume, and correlated it positively with the stored sperm. However, other methods, as described by Taylor et al. [31] and Gerofotis et al. [37], were used to assess the number of stored spermatozoa in the spermathecae. During copulation, the males transfer to the females not only their spermatozoa [38], but also other components with the seminal fluid [39]. It could be possible that the difference in spermatheca size between females that mated with W or AR males is due to either more seminal fluid and/or sperm being transferred during copulation by W than AR males. In order to clarify whether it is a matter of sperm production or transfer, future experiments are required to compare the sperm production between W and AR virgin males. Also, AR females had smaller spermathecae than the W females, irrespective of the type of male they mated with (*t*-test, $p = 0.004$). Spermathecae size in females and sperm quantity in males is also influenced by body size and weight, but we did not find statistical differences in the body weight of W and AR flies of the same sex. The size of spermathecae of AR females is not dependent on the type of male they mated with. AR females oviposited more eggs in the 10-day period before the measurement of their spermathecae, so we assume that their spermathecae volume was measured to be smaller than the W females because of storage depletion. In *Bactrocera tryoni*, the total number of stored sperm declines after the 15 days after mating, presumably as they are needed for fertilizations [40]. Methods like SIT/RIDL rely on the reproductive failure of W females after copulation with sterile or genetically modified males. A males' ability to transfer sperm to female spermathecae in order to switch off her receptiveness, prevent remating, and induce oviposition behavior are therefore crucial to the success of these methods [41]. Reproductive failure will be realized only if an adequate ejaculate (containing sufficient sperm and accessory gland products) is transferred from AR males to W females [42,43]. To our knowledge, there is no information about the internal structure of spermathecae and a female sperm selection process in *B. oleae*. Female olive fruit flies of are mostly monogamous [44,45], therefore, the possibility for sperm selection is very low.

Behavioral and physiological changes of mass-reared males, such as changes in mating initiation time and copulation duration, ability to join leks, courtship rituals, pheromone production, and attractiveness compared to wild fertile males can dramatically affect the copulatory success with wild population females [46]. Good mating characteristics of artificially reared males include high sexual competitiveness, photoperiod compatibility with the wild population, and ability to prevent remating in wild females. Studies with laboratory populations of *B. oleae* have indicated that mass-reared males can still possess these good mating characteristics, like the genetically modified OX3097D-Bol males [23]. Also, mating experiments between a laboratory-adapted hybrid population of the olive fruit fly and wild populations indicated the complete absence of mating barriers [11].

W and AR flies of the same population were used to compare their daily activity patterns through the detailed tracking of their locomotor activity with the Locomotor Activity Monitor (LAM25, Trikinetics, Waltham, MA, USA). Individual flies of sexually mature age were housed inside each of 32 tubes (25 mm diameter, 125 mm length) which were vertically crossed by infrared beams, so that when the fly moved inside the tube, it caused beam breaks. Counts of beam breaks were stored in a dedicated computer every minute and used to estimate the fly's locomotor activity levels, in addition to periods of inactivity. Personal unpublished data indicated that sexually mature AR males achieve lower locomotion levels during the day compared to the W male flies. During the night, W flies were inactive for longer time periods than AR flies, with the latter demonstrating a fragmented pattern of rest and activity. Such differences suggest that AR males might have difficulty in surviving and dispersing when released. Reduced rest time during the night might affect their fitness and daytime activities, such as courtship and mating in the field. Low locomotor activity levels may affect their dispersal and survival (Terzidou, Koveos, and Kouloussis, unpublished data). Field releases of irradiated olive fruit flies in arid regions demonstrated very low recapture rates, indicating low survival and dispersal rates [47].

In insect mass-rearing and production facilities, selection pressure is directed toward parameters that are important for high productivity, e.g., high fertility, high fecundity, short life cycle, longevity, and large size, but these parameters do not necessarily guarantee an optimal field performance. A high-quality insect in mass-rearing terms could easily be a poor performer in the field, especially since reproductive output and field performance might be traded off [48]. The success of genetic methods of population suppression depends greatly on the production of good-quality male insects, which are released into target wild populations. Quality parameters include the insect's ability to survive predators, interact with its environment, effectively forage for food and water, and locate, court, mate, and fertilize females of the target population [49]. Physiological costs of male–male rivalry interactions, courtship, and copulation may reduce the lifespan of males [37]. Thus, it is important that lab-produced flies are fit enough to survive and mate with as many females as possible of the wild population.

AR males used in this study achieved a similar proportion of mating with W females when competing with W males. However, considering the other quality parameters that are needed for achieving copulations once they are released in the field, their reduced locomotion might hinder their dispersal, or female courtship. But such problems have also been present in other mass-reared tephritid flies, they were effectively addressed, and several tools have been proposed to improve the mating performance of males used in SIT programs [50]. For instance, the exposure to the aroma of ginger root oil increased the mating competitiveness of adult sterile males of the Mediterranean fruit fly [51], and methoprene in the diet increased the sexual competitiveness of the Queensland fly [52]. An improved artificial diet of the olive fruit fly with microbiota increased adult survival under stress [53].

## 5. Conclusions

AR *B. oleae* males used in competitiveness experiments with W males managed to participate in similar percentages in formed couples with W females, although homotypic wild couples begun copulation earlier than heterotypic ones. Also, the spermathecae volume was larger in W females that mated with W males than the ones that mated with AR males. Less stored sperm may cause remating of W females, which is undesirable. A variety of other parameters, like locomotor activity, longevity, and photoperiod compatibility are considered with respect to the quality of AR insects, and there are many tools to improve it and achieve success in the genetic methods of wild population suppression, like Sterile Insect Technique, or Release of Insects carrying Domimant Lethal gene.

**Supplementary Materials:** The following supporting information can be downloaded at: https://www.mdpi.com/article/10.3390/crops2030018/s1; Figure S1: The beeswax dome placed on a glass base serving as oviposition substrate for the olive fruit flies. Inside the dome is a hydrated cotton ball to maintain high relative humidity; Figure S2: Artificial rearing of B. oleae larvae. Larva exiting the diet to pupate in the sand. Eggs are placed in contact with the diet and then covered with the Petri dish lid until hatching. After hatching, the lid is removed and the diet is kept moist by spraying on it with propionic acid solution (0.3 %) as needed; Table S1: Longevity (days) (SE) of wild and artificially reared adult flies.

**Author Contributions:** Conceptualization, A.T., D.K. and N.K.; methodology, A.T., D.K. and N.K.; formal analysis, A.T.; writing—original draft preparation, A.T.; writing—review and editing, D.K. and N.K.; visualization, A.T.; supervision, D.K. and N.K. All authors have read and agreed to the published version of the manuscript.

**Funding:** This research is co-financed by Greece and the European Union (European Social Fund-ESF) through the Operational Program Human Resources Development, Education and Lifelong Learning in the context of the project "Strengthening Human Resources Research Potential via Doctorate Research" (MIS-5000432), implemented by the State Scholarships Foundation (IKY).

**Institutional Review Board Statement:** Not applicable.

**Data Availability Statement:** The data that support the findings of this study are available from the corresponding author upon reasonable request.

**Acknowledgments:** We would like to thank J. Vontas, A. Kampouraki, I. Livadaras, and M. Konstantopoulou for providing the initial laboratory-adapted population of the olive fruit fly used in the experiments and for assisting in the establishment of the insect colony in our laboratory. We would also like to thank George Menexes for his assistance with statistics, and Eleni Yiacoumi for her comments on the original draft.

**Conflicts of Interest:** The authors declare no conflict of interest. The funders had no role in the design of the study; in the collection, analyses, or interpretation of data; in the writing of the manuscript, or in the decision to publish the results.

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
