# Peer review of "Mating Competition between Wild and Artificially Reared Olive Fruit Flies"

_2673-7655, doi:10.3390/crops2030018_

Round 1
Reviewer 1 Report
These are my main comments on the MS (crops-1790240) entitled : “Mating competition between wild and artificially reared olive fruit flies”.
It is a very interesting study dealing with the mating behavior of artificial and wild olive fly males. This knowledge will help us to enhance the use of SIT against fruit flies.
Generally, the study is well written. I did not locate any serious errors or flaws in Methodology, data presentation or discussion.
Authors may follow my suggestion to improve their MS.
Comments
Abstract. I would suggest adding some numerical data with the main conclusion.
Lines 145-146. Why did you choose t-test for these parameters and ANOVA for others? An explanation may be added here.
Lines 154-155.Please add an explanation about the number in parentheses. I imagine it is the statistical error of the mean.
Line 155. It is 43.9(5.5) in the text and 43.9(7.8) in the Table. Please correct it.
Line 176. Please correct to P<0.0001.
You have t - test described as “2-tailed” and “two-tailed”. Please keep the same terminology.
Lines 231-232. Are there any studies about mating behaviour of the olive fly in the field?
Discussion. It would be very interesting to compare these results with studies on other fruit flies.
Author Response
Please see the attachement

Reviewer 2 Report
This works aims to compare mating performances of wild and artificial reared olive flies. The manuscript is overall well written, the design is clear and methodology appropriate. I have some minor suggestions but also several major one to complete it, especially as there are some simple dissections that could be done to greatly improve the impact of this study.
minor changes
l98 : you say in the text that the successful mating pairs were removed, and a new male added, but what about the female ? was she also replaced by a W virgin one ? how did you maintained the 1:1 ratio ?
l104 : homogeneize the style, choose between 10 and ten everywhere
l123 : how did you observe eggs laid inside olives ? did you dissect them ? please give a little bit more precisions here
figure 1 : we miss the scale in the picture, and the legend miss the magnification degree and tool
statistics :
l143 why did you test for normality of your data ? this is the models residuals you are interested to test for this. You just need to look at your data distribution before running models.
l160-161 "There was no cage with fewer than 20 % of mated females which shows that the experimental conditions were appropriate and the tested insects were of the appropriate age and sexual maturation." would be more appropriate in the discussion
l168-172 I dont see the point of the table 1, as you give all your results in the text l152-154. I would suggest to remove it.
l160-166 you dont give the "n" associated with the average calculation, but you give it later in the 174-177 section. please add it everywhere.
table 2 : suggestion : put a box plot or a violin plot instead of the table, it will be way more digest
l126 it is not clear when exactly females were dissected : could you precise here whether this is just after the oviposition test ? so they all were aged of x+10 days ? it will be important for discussion
l246 please give latin name of medfly
please homogeneize the litterature list in the adequate format (refs 1, 5,14, 17, 27, 30, 35)
in the discussion, as you have an AR population old of more than 500 generation, would it be appropriate also to test for genetic clues of consanguinity ?
major revisions
the content of spermatheca is a mix of sperm but also seminal liquid, do you have refs that scientifically connect their number with spermatheca volume ?
The structure of those fly spermatheca seems quite complex to me, do you think that there could also be sperm selection by the female ?
I suggest you compare virgin female's spermatheca of each category for volume to confirm your hypothesis l 258-259
I also suggest for you to compare male sperm production (virgin adult ) of the 2 categories by dissection of male seminal gland to really know whether this is a matter of sperm production or transfer.
l289-292 you cite here personal unpublished data, you must give more protocolar precisions and stats as we have no clues here of how you obtained those results and their scientific soundness
Author Response
Please see the attachement

Reviewer 3 Report
The olive fly is a major pest of olives with capital relevance for the olive production industry. Among the different methods to limit the fly populations, releasing sterile or individuals carrying lethal genes in the field is of major importance and usefulness. However, the method success relies on ensuring the mating efficiency of the mass-reared modified adults. The authors conducted a series of laboratory assays to assess and compare the mating success of wild and artificially reared individuals. The manuscript is quite interesting and well written and data analysis is properly conducted. The results are efficiently discussed and, in my opinion, this is an interesting piece of research.
Minor remarks:
Line 30-31: Please consider writing an extra sentence depicting other methods of pest limitation (e.g. natural enemies).
Line 38-41: I miss a couple of sentences here with specific information on the survival of olive fly AR individuals compared to W individuals.
Line 60-67: This part should better described. Please provide coordinates for the “area of Thessaloniki”. The olives on which the flies were further allowed to lay eggs came from the same groves?
Section 2.1.: A couple of photos would significantly enrich this part of the manuscript (e.g. as supplementary material).
Line 107: “According to [13, 14]…”. Please write the reference and then the number (and hereafter if needed).
Lines 127-129: Please rephrase “Using the camera software, the volume of each spermatheca vesicle was measured and averaged (Figure 1).” since the volume was not measured using the camera/software but calculated subsequently.
Line 147: Please consider replacing a=0.05 with α=0.05.
Line 154 (and next paragraph): “55.5 (5.5) %”. Please specify what 5.5 stands for.
Table 2: If I´m not wrong, second row should read AR males.
Table 3: Please state the meaning of bolds in the table caption.
Author Response
Please see the attachement

Round 2
Reviewer 2 Report
Thank you for taking in account all the suggestions I made, I think that the paper is ready for publication now. Just one last comment : in the reference list there are still some formatting needed (ref 7 ), italic missing in ref 34 , ref 38, ref 45.